# Impact of ecotourism on abundance, diversity and activity patterns of medium-large terrestrial mammals at Brownsberg Nature Park, Suriname

**Dimitri A. Ouboter, Vanessa S. Kadosoe, Paul E. Ouboter**[ID]*

Institute for Neotropical Wildlife and Environmental Studies, Paramaribo, Suriname

* paul.ouboter@neowild.org

## Abstract

The impacts of ecotourism on biodiversity are poorly understood and the outcome of this type of research is often contradictory. On the one hand ecotourism could impact the occurrence, survival or behavior of species, on the other hand ecotourism is often mentioned as providing a "human shield" by deterring negative practices like gold mining, logging and hunting. Brownsberg Nature Park is easily the most visited protected area of Suriname, with a high number of ecotourists visiting from abroad. A four-year study on the impact of ecotourism on medium-large terrestrial mammals was carried out between 2013 and 2016 using 16 camera trap stations. The area has a clear gradient of tourism pressure, with the pressure decreasing further away from the lodging facilities. Evidently, the impacts of human presence on the mammal communities were more significant in the busiest areas. Most species avoided areas with many hikers or switched to a more nocturnal activity pattern. In these areas the impact was not reflected in species numbers, however it was causing a significant decrease in the diversity of mammals. On the other hand, vehicles had little impact on species avoidance or diversity, but did increase nocturnality, even more than hikers. A few species seemed to be "attracted" by hikers and/or traffic. Giant armadillos (*Priodontes maximus*) and spotted pacas (*Cuniculus paca*) used the pools in the road created by traffic. Ocelots (*Leopardus pardalis*), margays (*Leopardus wiedii*) and red-rumped agoutis (*Dasyprocta leporina*) seemed to favor human disturbance probably because of predator release. Some of the most impacted species were the jaguar (*Panthera onca*), puma (*Puma concolor*) and lowland tapir (*Tapirus terrestris*), all three species with significant contribution to ecosystem balance. Management measures should focus on lowering the number of hikers in popular places and limiting the number of vehicles in recreational areas.

**Data Availability Statement:** Data from 2013-2015 are available at GBIF (DOI: doi.org/10.15468/meceas). Data for 2016 are available at GBIF (DOI: doi.org/10.15468/apqeyf).

## Introduction

Ecotourism is generally regarded as beneficial for the protection of natural areas and is therefore expected to aid in the preservation of biodiversity [1, 2]. Local communities may also

**Funding:** This work was partly supported by funding from the Belgian Directorate-General for Development Cooperation (DGDC) and the Flemish Interuniversity Council (VLIR-UOS). The funders had no role in study design, data and analysis, decision to publish, or preparation of the manuscript.

**Competing interests:** The authors have declared that no competing interests exist.

benefit from the resulting economic turnover when involved in these projects, which can ultimately support a change from unsustainable resource extracting practices to sustainable conservation-oriented practices [3, 4]. While ecotourism is perceived to be a sustainable activity, the actual benefits and ecological consequences are poorly understood, and more research still needs to be done in this field [1]. Ecotourism plans for nature parks seldom take into account the potential negative impact from the very act of practicing ecotourism in said areas and should therefore include long-term monitoring and management of these effects [5, 6]. Ecotourism, which is defined as "responsible travel to natural areas that conserves the environment, sustains the well-being of the local people, and involves interpretation and education" [7], is certainly a commendable endeavor. However, to fulfill every aspect of this definition, is not an easy task and many ecotourism projects may fail in some of these promises. Studies have shown that involvement of the local community also helps in long-term sustainability of ecotourism in protected areas [2]. In the established exploitation of many protected areas, the involvement of local communities is overlooked (e.g. [8, 9]), and with that a failed comprehension of their significant contribution to environmental awareness and conservation [1]. Knowledge of the impact of tourist visitation on neotropical wildlife is still largely lacking. With an increasing trend in ecotourism, it is very important to determine any detrimental effects of tourism activities on the environment.

Results from studies so far, on the influence of ecotourism on biodiversity in the neotropics, have presented impacts that have ranged from little to noticeable negative effects [3, 6, 10, 11]. These differences may reflect contrasts in management strategies (e.g. max tourist density allowed) and ecosystem resilience, highlighting the importance of collaboration between ecologists and management authorities in creating sustainable ecotourism practices. Negative effects that have been documented includes increased stress, changes in behavior or disappearance from landscapes [4, 6, 10, 12]. Such changes can ultimately lead to decreased reproductive success in animals, lowering species abundance which can have cascading ecological effects, even leading to the extirpation of certain species. Community structures can change, giving rise to fluctuations in- and between species [13]. A decrease in abundance and diversity of species will be disadvantageous for local communities, the economy as well as for ecotourism. While the primary role of protected parks is the conservation of biodiversity and natural landscapes, which would be best achieved with minimal anthropogenic disturbances, the contradicting economic incentive for management entities to generate income for maintenance and development in the park often drives tourist numbers up. Consequently, the sustainable management of ecotourism ventures becomes a challenging endeavor. Infrastructure developments (e.g. roads, lodges) and park restrictions have to be carefully planned, aligned and managed to minimize the damage to the natural resources which the park is aiming to protect.

In Suriname, which has not been very tourism oriented in the past, ecotourism is an upcoming industry [14]. With 93% of the country covered by rainforests [15], most of which is pristine in the central, western and southern parts, and 14.5% of the land area being protected [16], the potential for ecotourism is remarkable. Suriname was the first country in the Western Hemisphere to issue nature preservation legislation. The hunting law of 1954 prohibits hunting of all mammals except game species, with restrictions to certain seasons and number of kills a person is allowed to make per species per hunting trip. In practice however, there is almost no control and regulation over hunting practices. Although local communities are allowed to hunt within protected areas, non-locals have been known to partake in this activity as well. There are also largely unregulated mining operations (both legal and illegal) occurring in different parts of the country, and even within protected areas [17]. Logging operations are well-regulated on paper, but environmental restriction and concession borders are often not observed. This presents a danger to the integrity of protected areas across the country.

Ecotourism can possibly help to shift from these destructive land uses to a more sustainable conservation-oriented practice. Here, we will be focusing on Brownsberg Nature Park (BNP), which experiences both of the above-mentioned pressures near its borders, as well as hosting ecotourism. Since the establishment of the park in 1970 [18], tourism has risen (largely uncontrolled) in BNP. It has become one of the most visited tourist sites in Suriname, with approximately 17,000 visitors in 2001 [5]. To experience the richness of the park, tourists have the option to travel by foot to several waterfalls, creeks, or scenic viewpoints. The trend herein is that the largest proportion of tourists chooses the least intensive journeys (i.e. the locations within the shortest distance from the lodging facilities). This creates variability in the amount of tourist pressure on different parts of the park and makes it ideal for an ecotourism impact study.

This study aimed to investigate the effects of ecotourism pressure on the presence, community composition, and behavior of medium-large terrestrial mammals by comparing photo captures in areas of the park with varying tourist and traffic pressure. This data was gathered during a 4-year continuous camera trapping monitoring study. We predicted that impacts on occurrence, diversity, and behavior of mammals would be more pronounced in areas with the highest tourist and traffic pressure. Ecotourism was shown to impact the behavior of many species displayed as avoidance of busy tourist routes or changing to a more nocturnal activity pattern. The Simpson's diversity index was inversely correlated to tourist numbers.

## Methods and study area

The Foundation for Nature Preservation in Suriname (STINASU) granted permission to carry out the research in Brownsberg Nature Park. No field permit number was given.

### Study area

Brownsberg Nature Park (5°01'N, 55°34'W) was established in 1970 as a protected rainforest area of approximately 12,200 hectares and is Suriname's only nature park [16]. It is situated northwest of the Brokopondo Reservoir, about 90 km south of the capital Paramaribo (Fig 1). Brownsberg is a ferro-bauxite capped mountain with a 470–530 m high plateau that stretches approximately 34 km in length and 13.5 km in width at its widest point [19]. The area is covered by humid forest and hosts a wide variety of habitats due to its wide range of elevations, steep slopes and gullies [5, 20]. This creates high diversity within a small area which is illustrated by the diverse fauna and flora, including endemic species which can also be found within the park [5]. Brownsberg Nature Park is home to at least 125 species of mammals consisting of "ten opossums, five pilosans, four armadillos, 58 bats, eight primates, 13 carnivores, five ungulates, and 22 rodents" [21]. This includes all of the felid and primate species known to occur in Suriname. Most of the surface area of Brownsberg is covered by mesophytic and meso-xerophytic rainforest. Other habitats occurring in the Park include xerophytic low forest, bamboo-liana forest, marshy streamside forest and swamp-marsh forest [19]. The climate is tropical, with two wet seasons (April/May to August and December to January) and two dry seasons (February to April and August to November/December) [19]. During the night and early morning precipitation is often enhanced by mist. Temperatures ranges from 19 to 30 °C and rainfall is approximately 1,985 mm/year [19]. In some years a very dry season is experienced with less than 60 mm of rainfall per month. During most of the year the eastern slopes and top of the mountain are moistened, both at night and in the early morning, by clouds and fog (pers. observ.). Most of the plateau and slopes within the park are in an almost pristine state. Small artifacts of Pre-Colombian Amerindians were found in the area [22], however, apart from a few bamboo thickets, nothing in the natural habitat reminds one of this history.

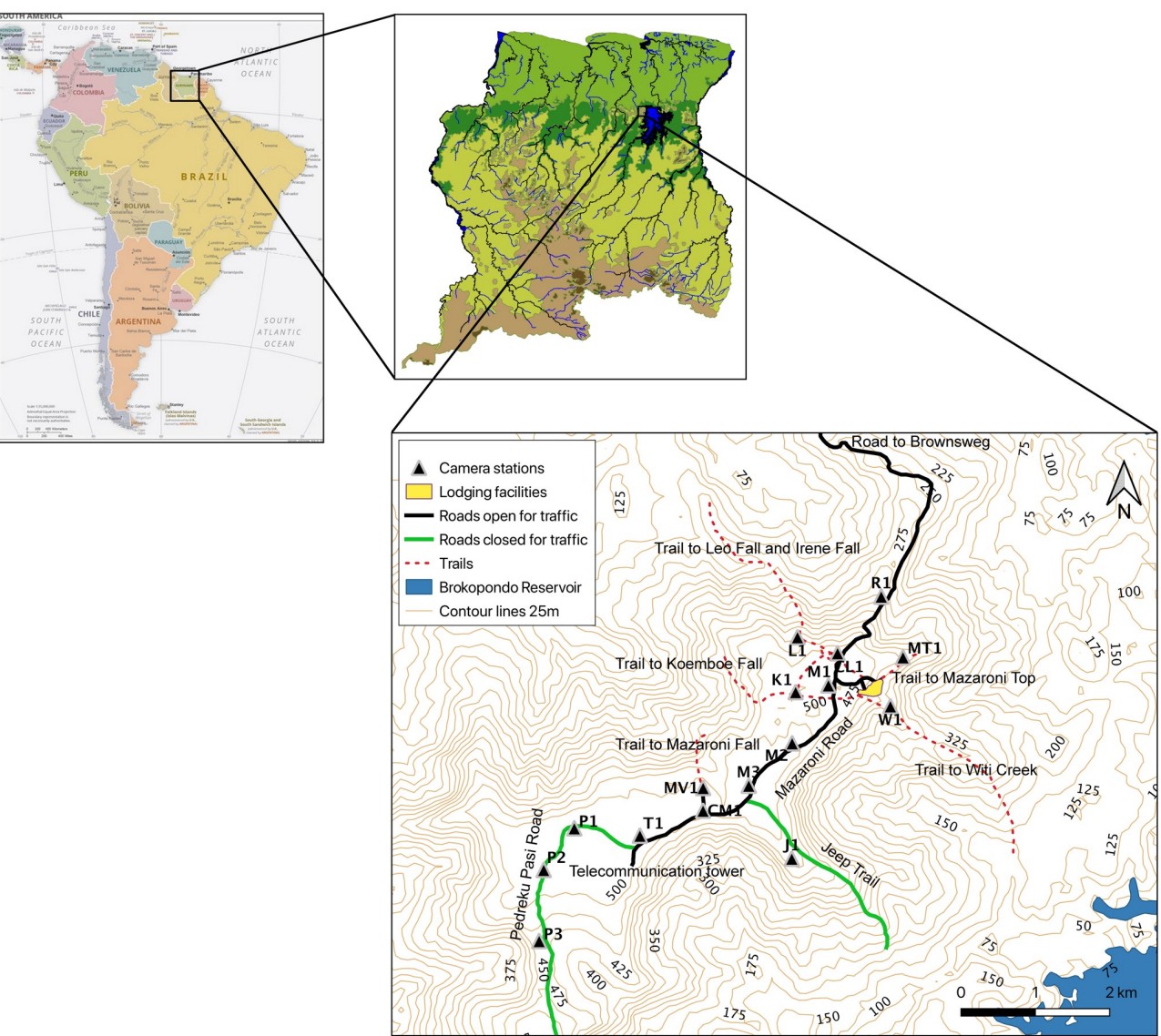

**Fig 1. Location of the study area and camera trap stations.**

From the late 17th century till 1964, Maroon (escaped slaves) villages occurred at the middle Suriname River, east of Brownsberg. According to De Dijn et al. [19] "The eastern Brownsberg foothills was likely the outer limits of areas used for slash-and-burn agriculture. . . . . . . . ., while the Brownsberg range itself was a tribal hunting ground". At the end of the 19th–and beginning of 20th century there was a short period of small-scale gold mining activities at the mountain. Some remnants of this period can still be found in some of the creek valleys. During the same period natural rubber was exploited in the forest of Brownsberg (locally known as balata bleeding) [19]. With the establishment of the Brokopondo Reservoir in 1964, the inhabitants of 11 Maroon villages of the middle Suriname River were translocated to Brownsweg, a settlement to the north of the Brownsberg mountain [23]. In 1970 the mountain received protection as a nature park [18]. This short history explains the relatively pristine condition of the more remote, southern part of the plateau and slopes, with hardly any recent hunting or logging, nor disturbance from tourism. However, disturbance at the foot of the mountain and on the

northern and eastern lower slopes is more pronounced. The gold rush has rekindled since approximately 1985 [24]. Gold mining operations have affected more than 661 ha of the park that is situated in the lowland [17]. Evidently, hunting and logging activities also occur in the lowland areas surrounding the mountain and at the lower slopes, mostly by inhabitants from the nearby villages of Brownsweg [5]. Part of the plateau of Brownsberg is still a bauxite concession of SURALCO (Surinam Aluminum Company), although no active bauxite mining has ever taken place. The study area is surrounded in three directions by "ecological barriers". To the North-West are gold mining operations, to the North is the Brownsweg village, to the East is the Brokopondo Lake, and to the South-East are more gold mining operations. However, there is a wide, relatively undisturbed corridor to the South-West.

Showing the road going up the mountain from Brownsweg village and continuing on the plateau as Mazaroni Road. Pedreku Pasi Road and Jeep Trail were closed for traffic. Included are trails leading to waterfalls, creeks and viewpoints. Map of South America copied from *The World Factbook 2021*. Washington, DC: Central Intelligence Agency, 2021. Political South America. https://www.cia.gov/the-world-factbook/.

Tourists can rent lodging facilities located on the northern part of the plateau (Fig 1), which can be reached through a road coming in from the North-East and starting at Brownsweg. This road continues as Mazaroni Road, the main road to the South-West of the plateau until it reaches the Telesur telecommunication tower. The other part of this road named Pedreku Pasi, which continues to the West and South on the central and southern part of the plateau, is closed for traffic (the PP Road locations in Fig 1). Another closed road splits off from the main road in southwestern direction and going downhill to the Witi Creek (Jeeptrail in Fig 1). Several hiking trails lead to waterfalls, creeks or viewpoints (Leo Fall, Koemboe Fall, Mazaroni Fall, Mazaroni Top and Witi Creek in Fig 1).

The northern part of the plateau with the lodging facilities, roads and several hiking trails is somewhat disturbed as a result of tourism activities. Disturbance decreases further away from the lodging facilities. The closed roads are considered to be in the most pristine parts of the park.

## Camera trap survey

The camera trapping study was conducted from December 2012 to December 2016. Mostly Reconyx PC900 cameras, containing a covert infrared flash, were used during the study. During 2012/2013 a few Reconyx PC600 cameras were also in use. Cameras were attached to trees between 30 to 80 cm (depending on the viewing angle from tree to trail) above the ground and secured with a Python lock. Cameras were set to take five rapid (<1 second interval) photos upon detection of a moving (warm) object, after which the camera had a delay of three minutes before arming again. A total of 16 camera stations with single or double camera setups were employed on different roads and trails (Fig 1). The double camera stations (one camera on each side of the trail or road) aided in capture maximization and provided photographs of both flanks of animals to assure individual identification of some species. Cameras were serviced each month to replace batteries, memory cards and desiccant, and for overall cleaning of the equipment.

## Data analysis

For the analysis, medium-large terrestrial mammals were defined as species with a bodyweight usually greater than 1 kg as an adult. This means that the smaller opossums (*Philander opossum*, *Metachirus nudicaudatus* and mouse opossums of the genus *Marmosa*), mice and rats

were excluded from the analysis. Also, principally arboreal mammal species, such as squirrels and monkeys, were excluded from the data analysis.

All photos were identified to either species of animal (using [25]) or classified as type of trigger by humans (e.g. tourist, vehicle, bicycle, hunter, gold miner). The species as opposed to the number of specimens was counted in each photo trigger (e.g. one peccary equals one trigger, but a group of ten peccaries also equals one trigger). When two or more consecutive triggers were of the same species, the first was counted and the second only after 30 minutes had passed since the first trigger. This measure was applied to prevent multiple counts of the same individual of several species that may linger in front of a camera for longer periods (agoutis, peccaries and armadillos) or may reappear after a short while (pumas). On the contrary, consecutive triggers by hikers or vehicles, were counted irrespective of time between triggers.

Common and scientific names are according to the IUCN Red List [26].

Human presence (hikers, vehicles, including noise and other related disturbances associated) usually has a negative impact on animals, and may cause avoidance which is reflected in a change in abundance, distribution, home range or activity pattern of species. However, human activities may also have a positive impact on some species by creating additional (usually open) habitats, pools, road connections, providing additional food and/or release from predation or competition. To evaluate the impact of tourist hikers or vehicles on the mammal community, we correlated tourist or traffic pressure, represented as number of triggers by hikers or vehicles, to species numbers and mammal diversity per camera station. Pearson correlation coefficient was used to evaluate the extent of correlation between explanatory variables. The species diversity was calculated using Simpson's Diversity Index [27], as being the most stable and easy to interpret diversity index [28]. The relative abundance index (RAI) was calculated for each species as the number of camera triggers per 100 trap days [29, 30]. Heatmaps were created using the Heatmap algorithm in QGIS 3.12.0 to visualize species RAI (root transformations were made for most species data because of the skewness of the data) in relation to disturbance. The impact of ecotourism on the mammal community was further investigated using Principal Component Analysis (PCA). The best groupings from PCA analysis were further evaluated with an Analysis of Similarity (ANOSIM) [31].

We used R version 3.5.1 for the analyses [32]. We compared activity patters of various mammal species between locations with undisturbed conditions and locations with many hikers or heavy traffic. In overlap (v0.3.2) [33] we fitted the activity patterns of species pairs and estimated the degree of overlap by providing a coefficient of overlap (delta). Delta ($\Delta$) is given as a number between 0 (indicating zero overlap in activity) and 1 (indicating complete overlap in activity). Following the recommendations of [34], we used the Dhat1 estimator when the smallest sample size was less than 50, otherwise we used the Dhat4 estimator. Since overlap is a descriptive statistic, we complemented it with Watson's $U^2$ test found in the package circular (v0.4–93) [35]. This non-parametric statistic is used to test for significant differences in two samples of circular data. The package suncalc (v0.5.0) [36] was used to extract sunlight phases for the study area. The percentage of diurnal, nocturnal and crepuscular activity was determined for each species in the different locations. Cathemeral activity was chosen as activity between astronomical dawn and sunrise and activity between sunset and astronomical dusk. Nine mammal species with primarily cathemeral, diurnal or crepuscular activity were chosen for comparison between areas.

In order to put "disturbance" into perspective, locations with on average more than 50 triggers by hikers in the busiest month of the year were included. "Heavy traffic" meant that in the busiest month of the year on average between 60 and 960 vehicles could trigger a camera. A trigger by a tourist could involve just one single tourist but also a group of several tourists.

**Table 1. Annual camera triggers by medium-large terrestrial mammals, tourists and traffic between 2013–2016.**

| Species | Common name | 2013 | 2014 | 2015 | 2016 | 4 years | RAI[a] |
|---|---|---|---|---|---|---|---|
| *Dasyprocta leporina* | Red-rumped agouti | 1,529 | 2,180 | 2,315 | 1,914 | 7,938 | 45.31 |
| *Puma concolor* | Puma | 419 | 377 | 405 | 387 | 1,588 | 9.06 |
| *Panthera onca* | Jaguar | 287 | 331 | 281 | 171 | 1,070 | 6.11 |
| *Leopardus pardalis* | Ocelot | 216 | 279 | 335 | 235 | 1,065 | 6.08 |
| *Mazama americana* | Red brocket | 127 | 140 | 90 | 205 | 562 | 3.21 |
| *Didelphis marsupialis* | Common opossum | 45 | 87 | 113 | 218 | 463 | 2.64 |
| *Cuniculus paca* | Spotted paca | 81 | 135 | 119 | 113 | 448 | 2.56 |
| *Myoprocta acouchy* | Red acouchi | 83 | 60 | 125 | 113 | 381 | 2.17 |
| *Mazama nemorivaga* | Amazonian brown brocket | 93 | 74 | 92 | 52 | 311 | 1.78 |
| *Dasypus kappleri* | Greater long-nosed armadillo | 13 | 81 | 118 | 94 | 306 | 1.75 |
| *Dasypus novemcinctus* | Nine-banded armadillo | 59 | 21 | 51 | 124 | 255 | 1.46 |
| *Tapirus terrestris* | Lowland tapir | 31 | 101 | 25 | 32 | 189 | 1.08 |
| *Leopardus wiedii* | Margay | 18 | 23 | 30 | 32 | 103 | 0.59 |
| *Priodontes maximus* | Giant armadillo | 10 | 21 | 38 | 30 | 99 | 0.57 |
| *Eira barbara* | Tayra | 14 | 26 | 18 | 24 | 82 | 0.47 |
| *Herpailurus yagouaroundi* | Jaguarundi | 5 | 8 | 23 | 16 | 52 | 0.30 |
| *Nasua nasua* | South American coati | 11 | 16 | 4 | 14 | 45 | 0.26 |
| *Pecari tajacu* | Collared peccary | 10 | 20 | 3 | 10 | 43 | 0.25 |
| *Tayassu pecari* | White-lipped peccary | - | 16 | 4 | 14 | 34 | 0.19 |
| *Myrmecophaga tridactyla* | Giant anteater | 1 | 3 | 3 | 3 | 10 | 0.06 |
| *Tamandua tetradactyla* | Southern tamandua | - | 4 | 3 | 2 | 9 | 0.05 |
| *Speothos venaticus* | Bush dog | 5 | - | 1 | - | 6 | 0.03 |
| *Procyon cancrivorus* | Crab-eating raccoon | - | - | 4 | - | 4 | 0.02 |
| *Didelphis imperfecta* | Guianan white-eared opossum | - | - | 2 | 1 | 3 | 0.02 |
| *Hydrochoerus hydrochaeris* | Capybara | - | 2 | - | 1 | 3 | 0.02 |
| *Cabassous unicinctus* | Southern naked-tailed armadillo | - | 1 | - | 2 | 3 | 0.02 |
| *Galictis vittata* | Greater grison | - | - | 1 | - | 1 | 0.01 |
| *Leopardus tigrinus* | Northern tiger cat | - | 1 | - | - | 1 | 0.01 |
| *Coendou prehensilis* | Brazilian porcupine | - | - | 1 | - | 1 | 0.01 |
|  | Tourists | 7,533 | 7,669 | 7,295 | 7,303 | 29,800 | 170.09 |
|  | Bicycles | 81 | 88 | 58 | 22 | 249 | 1.42 |
|  | Vehicles | 4,263 | 7,496 | 5,877 | 7,311 | 24,947 | 142.39 |

[a] RAI (Relative Abundance Index) is the number of triggers per 100 trap days.

## Results

The four years of camera trapping corresponded with 17,520 trap days, during which the cameras were triggered 70,071 times by an animal, human or car. Tourists caused the highest number of camera triggers (42%), followed by vehicles (36%) and only 22% of the triggers was by a medium-large terrestrial mammal. Most triggers by mammals were caused by red-rumped agoutis (RAI 45.31), pumas (RAI 9.06), jaguars (RAI 6.11) and ocelots (RAI 6.08) (Table 1). A total of 29 species of medium-large terrestrial mammals were photographed. Nine species (31%) were photographed only after the first year, and four (14%) only after the second year. From the third year on, no new species were added up till 2020 (unpubl. data).

The Pearson correlation between the number of species and tourists or cars was not significant. It would otherwise be expected that an increase in disturbance would result in a decrease

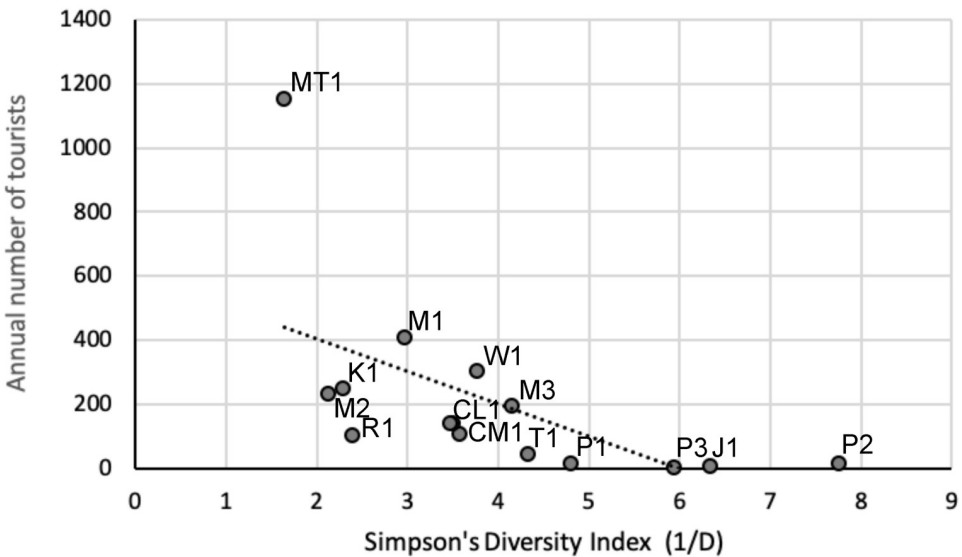

**Fig 2. Impact of the annual number of tourist triggers per camera station on the Simpson's Diversity Index (1/D).**

in the number of species. On the contrary, the second busiest trail, to Mazaroni Top, had the highest number of species. However, there was a significant negative correlation between the annual number of tourists and Simpson's Diversity Index (r = -0.622, p = 0.01) (Fig 2) and a significant positive correlation between the distance to the tourist center and Simpson's Diversity Index (r = 0.709, p = 0.002) (Fig 3).

The Pearson correlation between the number of triggers of a mammal species and the number of tourists was significantly positive for the margay, ocelot and spotted paca and significantly negative for the lowland tapir, jaguar and collared peccary (Fig 4). The correlation

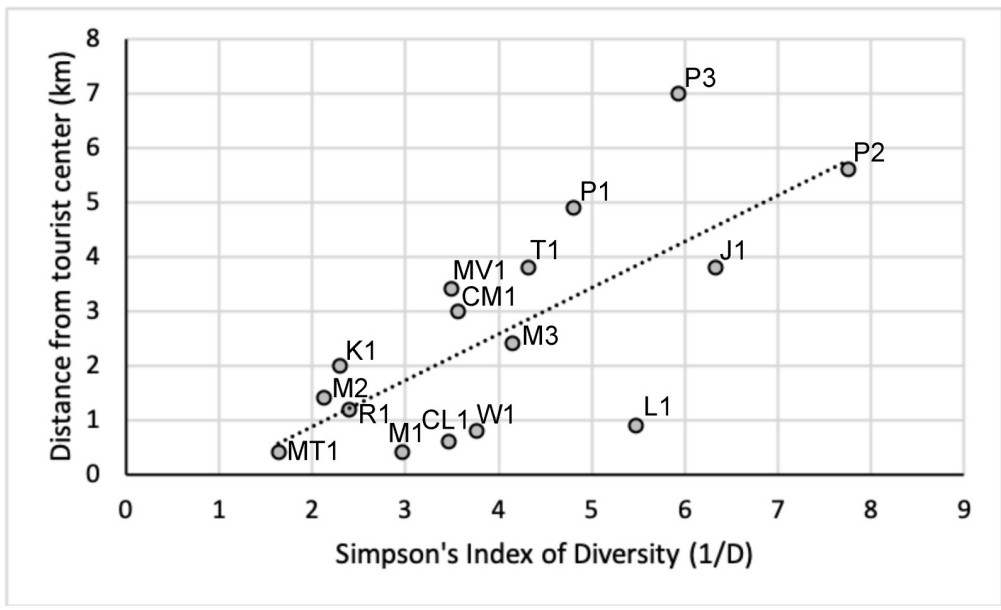

**Fig 3. Correlation between the distance of a camera station to the tourist center (in km) and the Simpson's Diversity Index (1/D).**

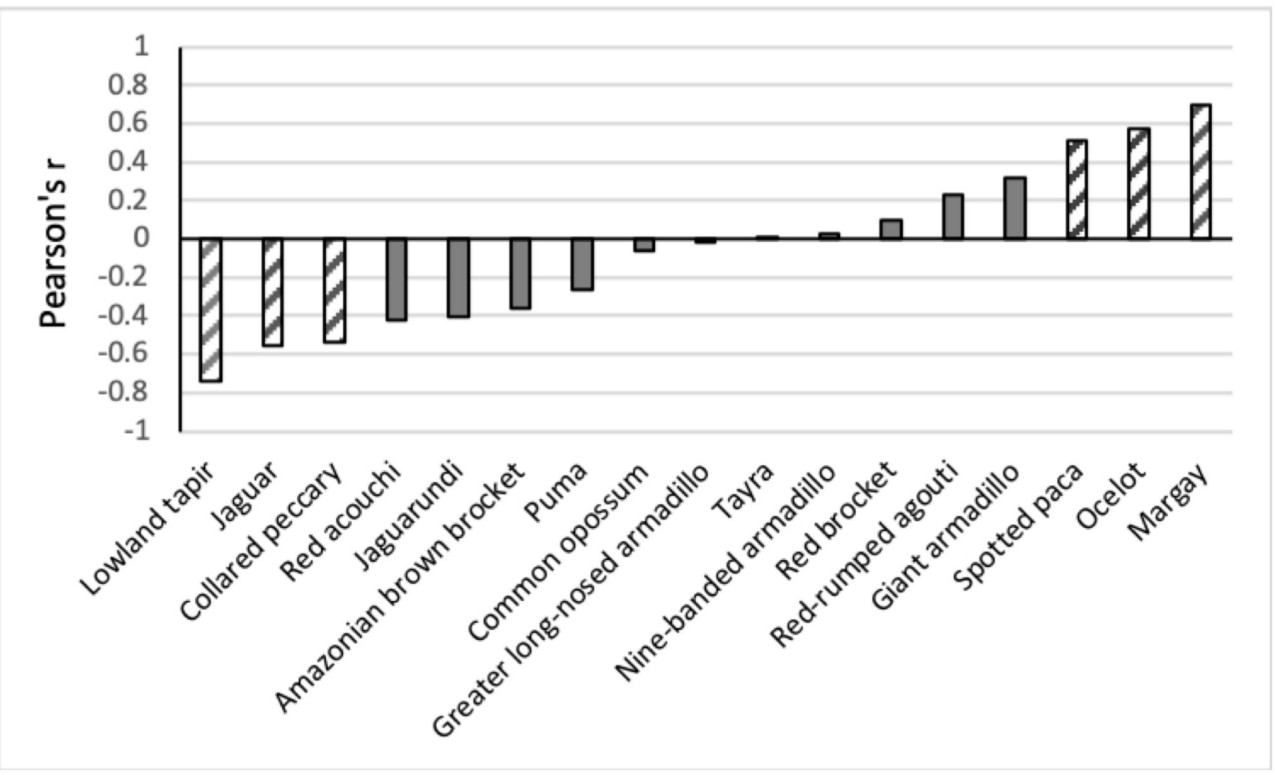

**Fig 4. Pearson correlation (r) between the annual number of tourist triggers per camera site and various mammal species.** Solid gray—r is not significant (p>0.05); striped-gray bars—r is significant (p<0.05).

between individual species and the number of cars was significantly positive for the spotted paca (r = 0.503, p = 0.047) and negative for collared peccary (r = -0.649, p = 0.006).

The heat maps in Fig 5 illustrate the impacts of the significant correlation between species and tourists and/or cars. The top maps of Fig 5 illustrate that Leoval and Mazaronitop are the most frequented sites by tourists. Significantly fewer tourists hiked to the other more remote trails of the park or continued on the Mazaroni Road to the south of the plateau. Hardly any tourist entered the roads that are permanently closed for traffic. Most drivers, after reaching the plateau, are inclined to leave their vehicles at the parking lot near the lodging facilities. This explains the limited number of vehicles that continued on the Mazaroni Road to the South. However, a greater amount of traffic that still occurs in this part of the plateau can be attributed to the telecommunication company (Telesur) which performs frequent maintenance services at their communication tower. The only two vehicles that were ever registered on the closed roads were related to a one-time visit by law-enforcement personnel on All-Terrain Vehicles.

According to Fig 5, collared peccaries avoided both hikers and traffic and were mostly recorded on closed roads and the trail to Witi Creek. Tapirs mostly avoided hikers and were mainly seen on the closed roads. However, on several occasions they were recorded from localities with limited traffic, especially towards the end of Mazaroni Road. Jaguars seemed to be mostly following the wider roads, and this also included roads with traffic. However, heavy traffic pressure caused a change in their behavior from a cathemeral to a more nocturnal activity pattern (see below). Jaguars appeared to avoid narrow trails with a high frequency of hikers. On the contrary, positive correlations were found for the margay, ocelot, and paca: Margays

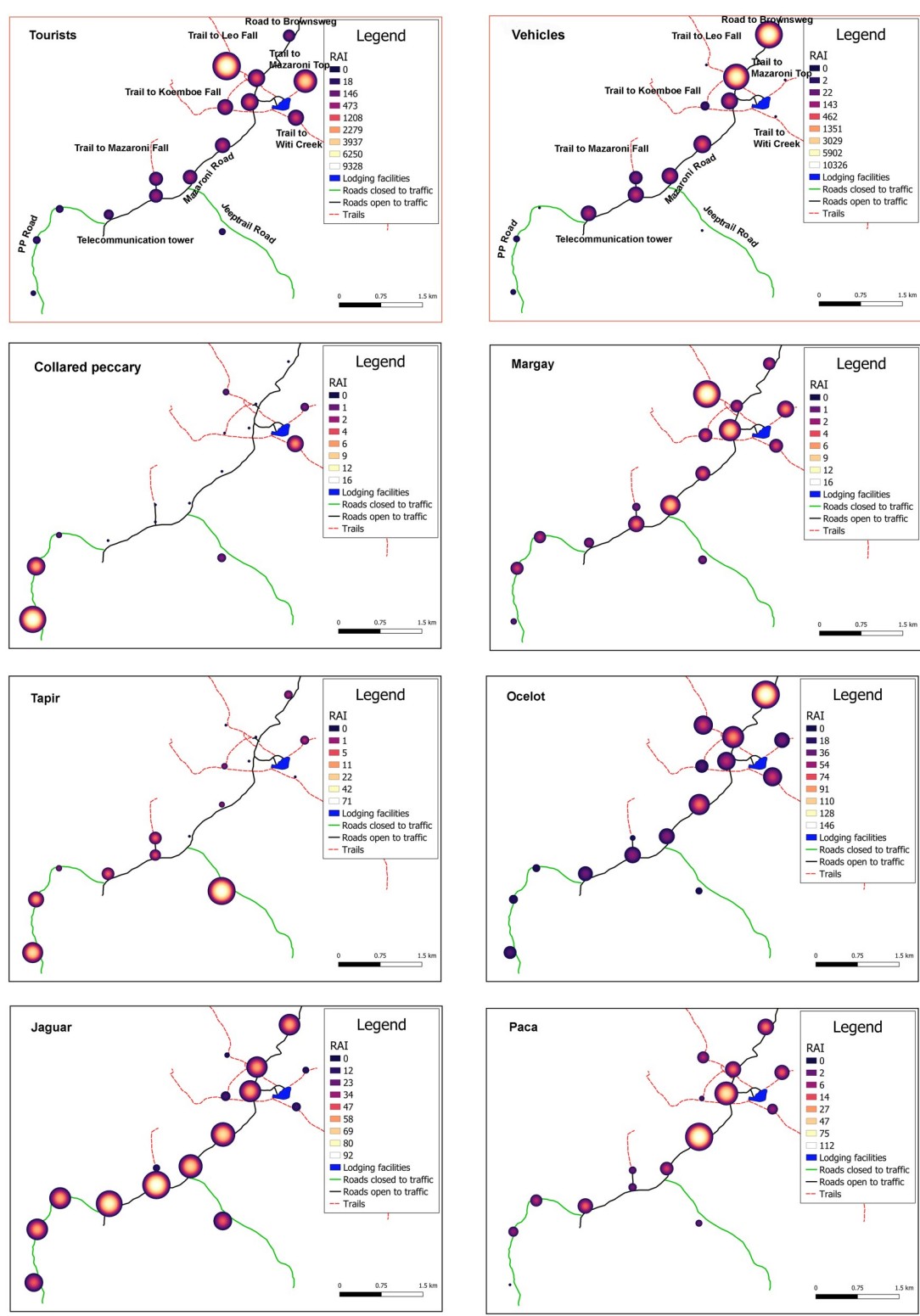

**Fig 5. Heat maps with red borders indicating disturbances (hikers and traffic) and heat maps with black borders indicating species.** Negatively impacted species on the left, positively impacted species on the right. RAI—relative abundance index is displayed as number of triggers per 1,000 trap days for better visualization. Roads, trails and lodging facilities were mapped by the authors. Maps were created in QGIS.

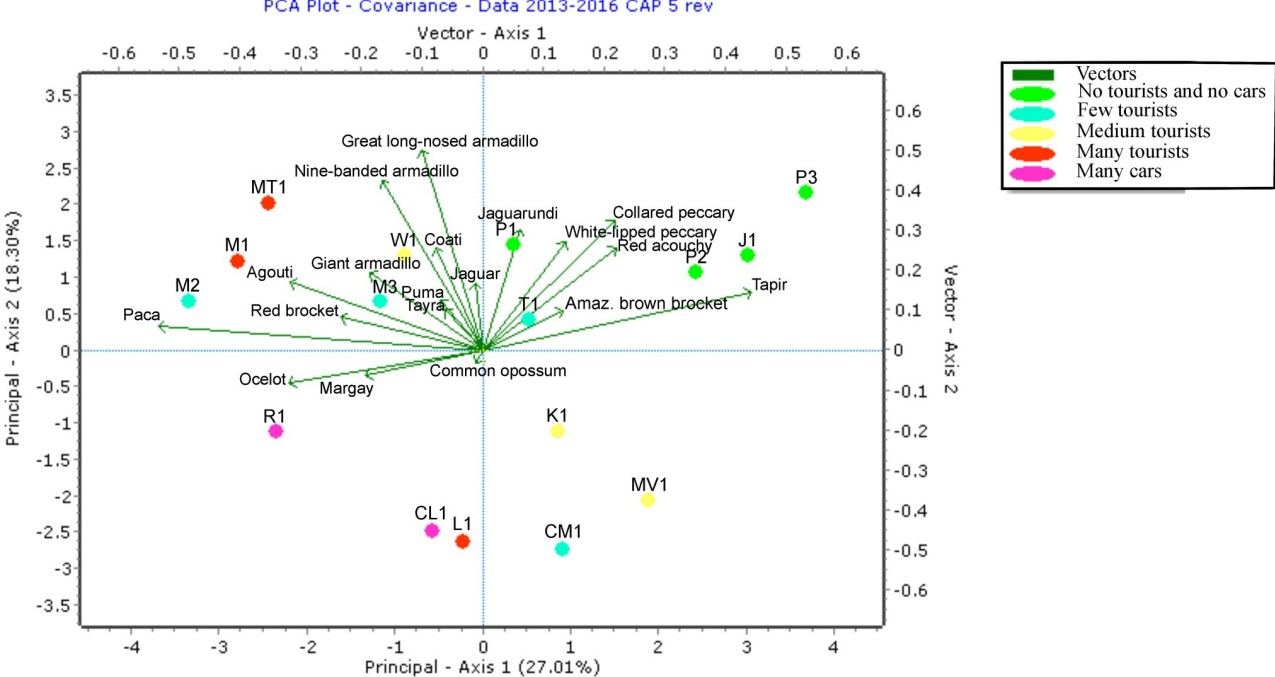

**Fig 6. PCA plot of camera stations with species grouped according to level of disturbance.**

were often recorded at localities with many tourists (at night), for ocelots, this was especially the case at roads, while pacas seemed to be most abundant at camera stations near pools in the road.

The Principal Component Analysis graph (Fig 6) shows that the closed roads, i.e. the undisturbed locations, were grouped in the right upper corner. Species with a preference for this area were tapir, red acouchi, collared peccary, white-lipped peccary, jaguarundi and Amazonian brown brocket. These were also the species with a negative correlation with tourist numbers shown in Fig 4. The more disturbed locations are grouped to the left side of axis 1 and the species that triggered the cameras more often in these areas are the margay, ocelot, paca, red brocket, red-rumped agouti and giant armadillo. These are the same species that showed a positive correlation with tourist numbers (see Fig 4). The other species of armadillos and South American coati are projected in between these two areas. ANOSIM results for all groups are significant (r = 0.275, p = 0.002). Pairwise tests between groups were only significant between "Many tourists" and "No tourists and cars" (r = 0.611, p = 0.028) and between "Few tourists" and "No tourists and cars" r = 0.396, p = 0.014).

Besides the impact of the number of tourists and cars on the occurrence/occupancy of several individual mammal species and diversity of the mammal community in general, disturbances may likely have an impact on the behavior of mammals by increasing avoidance of people and traffic. Hiking and driving on the roads and trails are, for more than 90%, diurnal activities. For species that are cathemeral, diurnal, or crepuscular, avoidance of human activities may be accomplished by a shift in their activity pattern.

Shifts in activity patterns due to disturbance by hikers and vehicles are statistically significant for the jaguar and puma (Fig 7). Both species had a shift to more nocturnal activity, and this was rather noticeable due to traffic rather than hikers. The shift to more nocturnal activity for the ocelot was solely due to the significant impact of hikers.

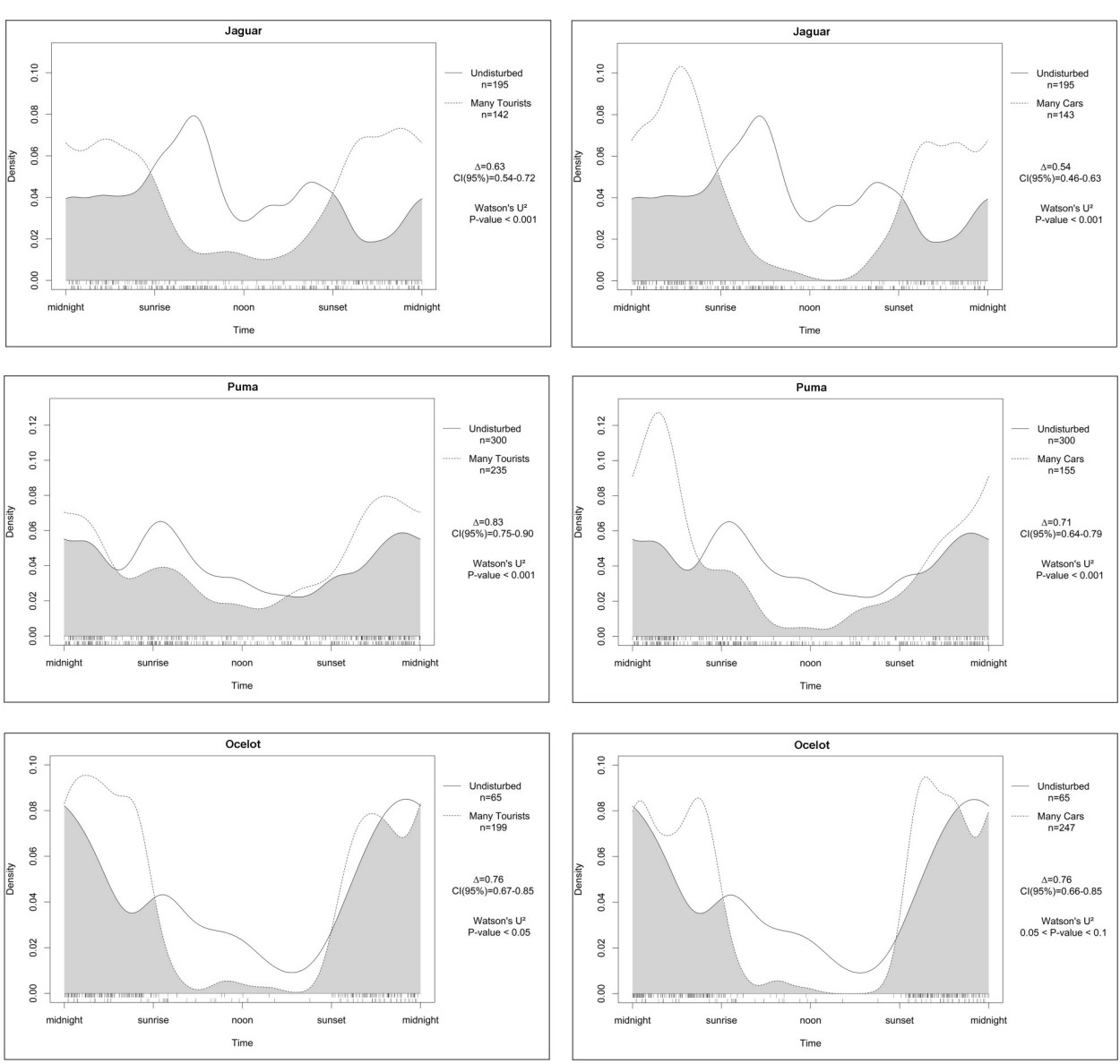

**Fig 7. Differences in activity patterns of several cathemeral and nocturnal predators between undisturbed areas and areas with a high number of tourists (left) and many vehicles (right).**

Shifts in the activity of diurnal non-predatory species, due to disturbance, were usually not significant and resulted mostly in less activity in the late afternoon and an increase in activity in the early morning (Amazonian brown brocket, red acouchi, red-rumped agouti) (Fig 8). This was in contrast to the cat species. Traffic caused a shift in the activity pattern of the nine-banded armadillo to more activity in the early night.

Changes in activity patterns due to human disturbance are further illustrated in Fig 9.

The impacts of human disturbance were most noticeable on jaguars, changing their 59% diurnal activity to 20% as a result of hikers, and only 11% remaining with heavy traffic. Pumas followed the same pattern although those shifts are less pronounced. Ocelots are usually more nocturnal than the larger cats, but nonetheless, their 28% diurnal activity in undisturbed areas, diminished to 4–5% in disturbed areas. Amazonian brown brockets avoided the presence of

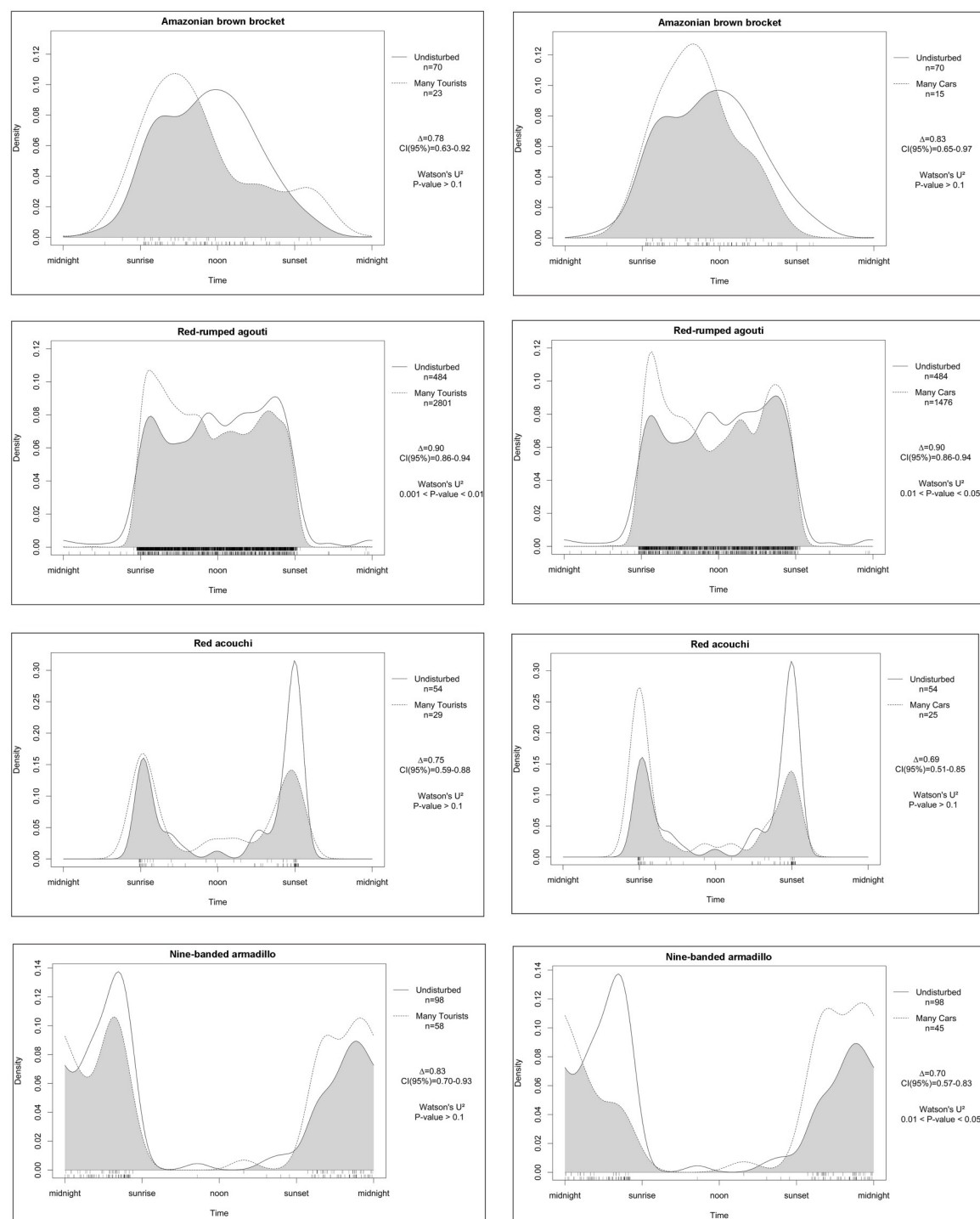

**Fig 8. Differences in activity patterns of several diurnal, cathemeral and nocturnal non-predator species between undisturbed areas and areas with a high number of tourists (left) and many vehicles (right).**

hikers with a slight shift to more crepuscular and nocturnal activity (18% less diurnal activity). Traffic did not seem to have an impact on their diurnal activity. Red acouchi on the contrary became more crepuscular due to traffic, with a shift of 12%. No effect was seen in the other species.

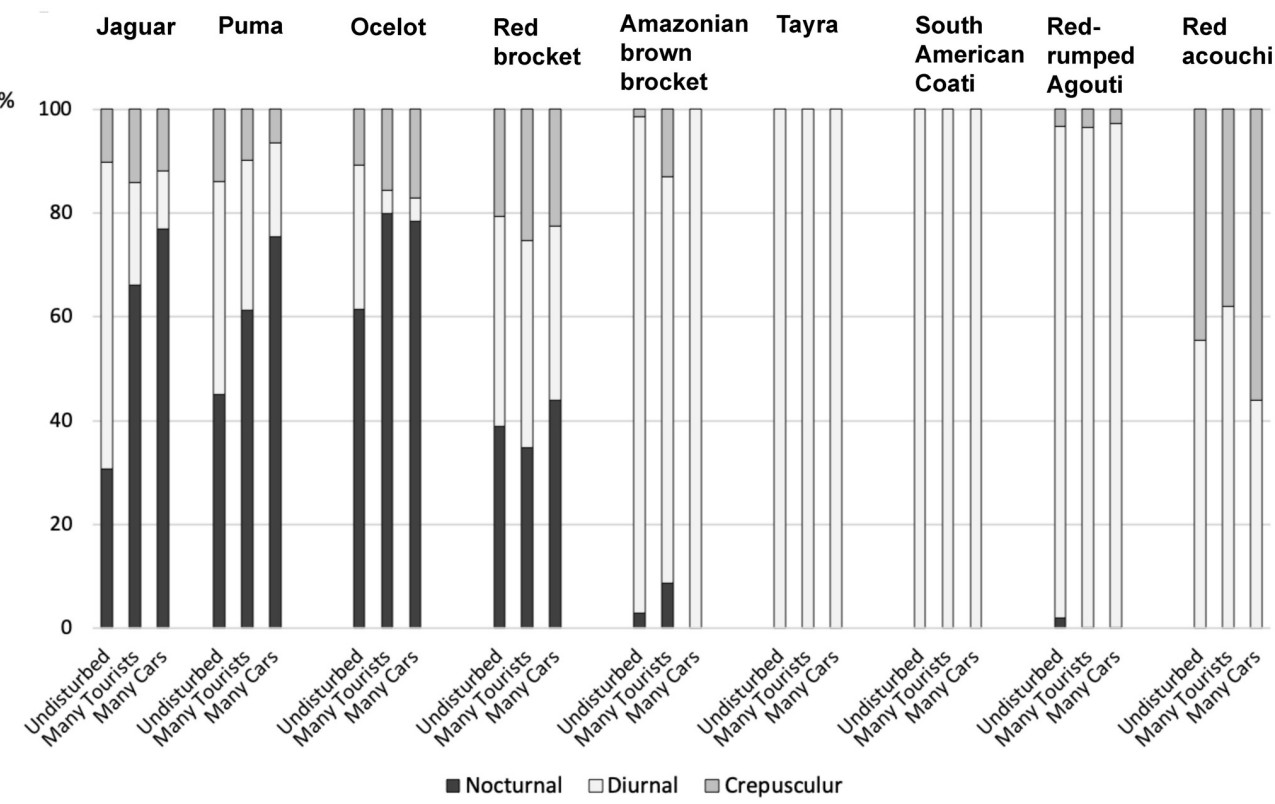

**Fig 9. Activity patterns of several mammal species divided into nocturnal, diurnal and crepuscular percentages, compared between undisturbed sites and sites with many tourist hikers or vehicles.**

## Discussion

In a comparison of the mammal communities of seven tropical rainforest sites, the Central Suriname Nature Reserve came out as the richest with 28 species (including small mammals too) [37]. This illustrates that Brownsberg Nature Park, with 29 medium-large terrestrial mammal species, is one of the richest areas in the tropics.

Camera triggers (RAI) were very high for the red-rumped agouti (45.31), puma (9.06), and jaguar (6.11), even higher than the numbers reported for the period 2013–2015 for the same area [38]. These are the highest number of camera triggers (per 100 trap days) reported for agouti [39–45] and puma [39, 40, 46–48], and one of the highest reported for jaguar (see overview in [49]). The density estimates for the jaguar population of Brownsberg, calculated from a nine-year study was also relatively high, varying between 0.51 and 4.21 individuals/100km$^2$ [50].

Camera trapping is often mentioned as an excellent method to study elusive and rare species (e.g. [51]). Our results support the efficiency of the camera trapping method for studying relatively abundant elusive species, however not for rare species. Only 69% (20 species out of 29) of species present in the area were photographed during the first year, and 14% of species (4) were recorded only in the third year.

The results of this study highlighted that (eco)tourism had an impact on the mammal community of Brownsberg Nature Park. An increase in the number of tourists on the road and trails resulted in a decline in the mammal diversity. Several species seemed to avoid areas with noticeable human activity, especially lowland tapir, jaguar, collared peccary, red acouchi,

jaguarundi, Amazonian brown brocket and puma. Avoidance of areas with high human traffic (in this case researchers) was also reported for several large mammals in a rainforest area at Gunung Leuser National Park (Sumatra, Indonesia) [52]. A similar study at the Tiputini Biodiversity Station in Ecuador [53] examined the correlation between human traffic on rainforest trails and mammal functional groups. They only found human traffic to have a (negative) effect on ungulates. They recorded the strongest negative effect for the white-lipped peccary and lowland tapir, whereas a weaker correlation with red brocket and collared peccary was noted. The Amazonian brown brocket had too low numbers to be analyzed separately [53]. The results from our study showed a similar negative correlation with all ungulates except the red brocket, which showed a not significant positive correlation. Similarly, the Amazonian brown brocket showed a negative correlation with the number of hikers. The activity pattern of the Amazonian brown brocket appeared to be diurnal versus a cathemeral activity pattern for the red brocket, which may explain the difference in correlation. Red brockets have been visually observed during our study near the tourist lodges by tourists and by the research team, on some occasions even with a juvenile in tow. This may represent a true example of spatial refuge of prey in spaces where predator displacement occurs due to human activity, although the effect is not strong enough to show statistical significance. The red brocket may show more behavioral plasticity, which may also explain the different results found in [53]. The study of [53] was only carried out from January to March (60 days) for three years, but nevertheless it could be interesting to investigate a possible seasonal component to the effect. Red brockets in both areas have been shown to vary in activity pattern throughout the year [38, 54]. This may in turn be linked to diet or reproductive activity which has also been found to vary throughout the year [55, 56]).

The margay, ocelot, and spotted paca showed a significant positive correlation with the number of hikers. This correlation was also positive for the giant armadillo and red-rumped agouti. We assumed that the paca and giant armadillo were most likely attracted to the pools in parts of the road, which had otherwise heavy traffic. These two species showed a significant affinity with the pools in parts of the roads, a behavior regularly captured on photographs primarily at night.

The explanation for a positive correlation between the high numbers of hikers and the margay, ocelot, and red-rumped agouti can be sought in a possible partial release of potential predation or competition by the jaguar and puma, which both slightly avoid areas with high human disturbance. Jaguars and pumas are known to kill smaller cats [57], and it was even reported that ocelots may represent a significant portion of the diet of jaguars during the wet season in the Talamanca mountains of Costa Rica [58]. Release of predation was also reported by [59] who performed a mammal community analysis and discovered some spatial separation of predators and prey, with prey more likely to occur around humans. While margays typically prey on small mammals, ocelots regularly prey on medium-sized mammals [60, 61]. On Barro Colorado Island in Panama, agoutis *(Dasyprocta punctata)* are a significant prey item for ocelots [62, 63]. Competitive release from jaguars creates a shift in the diet of ocelots, allowing ocelots to take larger prey than in areas where jaguars still occur [64]. Although agoutis can be preyed upon by jaguars, pumas, and ocelots alike, they are more likely to survive attacks by ocelots (e.g. [62]). Hence, the effect for ocelots may be two-fold, on the one hand by the absence of jaguars and pumas and on the other by attraction to agoutis.

Although there seemed to be little impact of traffic on species numbers and community diversity, traffic still had the highest impact on the activity pattern of the three largest cats. It resulted in a shift in their activity pattern from daytime to mostly nighttime activity.

Shifts in mammal activity were also noted due to the presence of hikers, mostly shifting parts of their diurnal activity to nocturnal (cats) or concentrating diurnal activity primarily in

the morning (Amazonian brown brocket and red-rumped agouti). The shift from afternoon activity to mainly morning activity can probably be explained by a higher tourist activity in the afternoon hours compared to the morning. The shift from early morning to evening activity in the nine-banded armadillo as a result of heavy traffic, may very well be a secondary effect caused by the shift of jaguar and puma activity to mainly early morning.

Shifts in activity pattern were also reported by [65] for the cheetah in Amboseli National Park (Kenya), by becoming more crepuscular in an avoidance strategy in response to tourists. In Sumatra, the sun bear and tiger changed their activity pattern from mostly diurnal to nocturnal in areas of high human traffic [52]. Seventy-six studies on the human impact on daily activity patterns of animals were analyzed by [66] and they concluded that there was a significant increase in nocturnality in response to human disturbance. Seven studies were from tropical South America, however, none of these involved the effects of tourism. The temporary closure of a highly visited national park in Thailand resulted in leopards becoming more diurnal in the absence of tourist disturbance [67]. A study in the Atlantic Forest in Argentina/Brazil reported increased nocturnality in pumas in disturbed areas [68].

Nocturnal mammals may experience less of a hindrance than diurnal species because most tourist activity on the trails happens in the daytime. Therefore, it was expected that diurnal mammals would primarily be affected to a greater extent by tourism than nocturnal mammals. This was true for the collared peccary, red acouchi, jaguarundi, and Amazonian brown brocket. The only diurnal species that did not experience a negative impact from tourism-related disturbance were the tayra, South American coati, and red-rumped agouti. However, cathemeral species were also impacted: jaguar, puma and lowland tapir. Several mostly nocturnal species were indeed hardly impacted or even experienced a positive correlation with tourist disturbance: margay, ocelot, spotted paca and giant armadillo.

Jaguars are usually reported as being nocturnal over most of their range [69–73]. Only in the relatively open wetland habitat of the Pantanal (Brazil) [74, 75] and in Eastern Ecuador [54, 76], jaguars are reported as mostly or more diurnal. In the undisturbed area of Brownsberg, the Jaguar is more diurnal (59%) than nocturnal (31%). The central and southern parts of the plateau and slopes of Brownsberg Nature Park are considered to be relatively pristine among tropical rainforest areas in the world. It is therefore assumed that this observed activity pattern of the jaguar also resonates with their original activity patterns in rainforest habitat in other parts of the jaguar's range before it was impacted by human disturbance. This assumption is supported by [76], who showed that in the Yasuní Biosphere Reserve in Eastern Ecuador, a relatively undisturbed lowland rainforest area, male jaguars were diurnal in 71% of the triggers and females in 52%. Additionally, in the undisturbed area of Brownsberg, the Puma was also cathemeral (41% diurnal, 45% nocturnal).

Human activities in protected areas increase the chance of encounters with large carnivores and this may harm the survival of sensitive species [77]. Disturbing predators during hunting or feeding may influence the effectiveness of their predation efforts and thereby impacting predator survival [78]. Large mammals in particular act as keystone species that can maintain diversity and balance within habitats and are therefore used as indicator species for the health of certain ecosystems [79]. The jaguar (*Panthera onca*) an important keystone species, fulfills a critical role in balancing the rainforest faunal community, simply through its wide variety diet consisting of approximately 85 species of animals [80]. The lowland tapir is considered the only species in the Amazon region that can disperse large seeds over long distances [81]. Tapirs disperse at least 39 species of plants and improve the germination of seeds. They generate a unique seed dispersal pattern in the forest [82]. Losing tapirs in an area will therefore change the vegetation dramatically. Peccaries are considered "ecosystem engineers" [83], because they have an enormous impact on the germination and distribution of many palm species.

Imbalance in the mammal community can thus have varying impacts on the forest ecosystem through complex cascading effects of predator-prey and consumer-plant dynamics [84, 85]. There are many examples of the effects of partial defaunation on ecosystems, whereby remaining species may [86] or may not [87, 88] be able to fill in the functional gaps created by defaunation. There is even evidence that the plant community may decreases in diversity [89].

In-depth research and prolonged monitoring are required to evaluate if cascading impacts have already changed the rainforest ecosystem where most tourist activities have occurred in Brownsberg Nature Park.

When ecotourism management is performed correctly, this practice can be beneficial for both humans as well as for wildlife communities. Researchers [90] in the Lapa Rios Ecolodge Natural Reserve in Costa Rica have argued that ecotourism has benefitted the wildlife community in this area, by acting as a "human shield" and deterring negative practices such as gold mining, logging, and hunting. The same may very well be evident for Brownsberg Nature Park, where miners, loggers, and hunters seemed to mostly limit their activities to the foot and lower slopes of the mountain, where hardly any tourist trails exist. There have been observations of miners attempting to terminate existing trails (by intentionally creating forest fires) to Witi Creek in order to dissuade tourists from visiting and creating for themselves release of being caught in their illegal activities. Another example was observed after the park was closed at the onset of the COVID-19 crisis in 2020, during which an increased number of miners/hunters was seen in the park (unpublished data).

Considering the negative impacts of hikers and traffic on the mammal community, especially at certain busy sites, managing authorities might consider applying measures to reduce these impacts. Traffic has had the most impact on the road coming up the mountain and continuing to the lodging facilities, while hikers had the most impact on the trails going to the nearest waterfalls (Leo/Irene Falls) and viewpoint (Mazaroni Top). Based on the evaluation of the extent and magnitude of tourist presence in the different areas, the following management measures can be proposed to minimize the pressure on wildlife: 1) limiting the total number of tourists to the park per visitation; 2) diverting tourists as much as possible to other trails; 3) limiting access to the park for private vehicles; 4) limiting the access to the park for vehicles beyond the reception of the park; 5. Creating activities at the lodges to divert pressure from the trails (e.g. educational activities, cultural performances, etc.). Measures 1 and 3 may be quite difficult to implement and may have serious financial consequences for the managing authority STINASU. Diverting tourists to other trails to relieve pressure on the busiest ones (measure 2) can also have dubious effects since this will only increase the area that will ultimately be exposed. Measure 4, limiting access for vehicles beyond the reception, could easily be implemented, and would drastically diminish disturbance at the southern part of the park. However, this would not differ for the busiest areas. Measure 5 could possibly relieve a little bit of pressure on the most impacted trails.

## Acknowledgments

We thank all persons that on occasion helped us in the field: Geeta Thakoerdien, Usha Satnarain, Niradj Hanoeman, Mercedes Hardjoprajitno, Cindyrella Kasanpawiro, Richero Kasanwidjojo, Carlo Koorndijk, Remesa Lang, Gunovaino Marjanom, Devina Bissumbhar, Naomi Reussien, Ashvin Sewsahai, Artie Sewdien and Sergio Doelasan.

We further thank the (former) coordinator of the VLIR MSc Sustainable Management of Natural Resources, Riad Nurmohamed, and the VLIR Secretariat for providing administrative support.

## Author Contributions

**Conceptualization:** Paul E. Ouboter.

**Data curation:** Dimitri A. Ouboter, Vanessa S. Kadosoe, Paul E. Ouboter.

**Formal analysis:** Dimitri A. Ouboter, Paul E. Ouboter.

**Funding acquisition:** Paul E. Ouboter.

**Investigation:** Dimitri A. Ouboter, Vanessa S. Kadosoe, Paul E. Ouboter.

**Methodology:** Vanessa S. Kadosoe, Paul E. Ouboter.

**Project administration:** Vanessa S. Kadosoe, Paul E. Ouboter.

**Resources:** Paul E. Ouboter.

**Supervision:** Vanessa S. Kadosoe, Paul E. Ouboter.

**Validation:** Dimitri A. Ouboter, Vanessa S. Kadosoe, Paul E. Ouboter.

**Visualization:** Dimitri A. Ouboter, Paul E. Ouboter.

**Writing – original draft:** Dimitri A. Ouboter, Paul E. Ouboter.

**Writing – review & editing:** Dimitri A. Ouboter, Vanessa S. Kadosoe, Paul E. Ouboter.

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
