## [Decision Letter · Decision Letter 0]

23 Apr 2021

PONE-D-21-10214

Impact of ecotourism on abundance, diversity and activity pattern of medium-large terrestrial mammals at Brownsberg Nature Park, Suriname

PLOS ONE

Dear Dr. Ouboter,

Thank you for submitting your manuscript to PLOS ONE. After careful consideration, we feel that it has merit but does not fully meet PLOS ONE’s publication criteria as it currently stands. Therefore, we invite you to submit a revised version of the manuscript that addresses the points raised during the review process.

There was one referee of your paper who recommended publication after minor revision based on his/her comments. I completely agree with this view and look forward to seeing the updated version for further consideration.

We look forward to receiving your revised manuscript.

Kind regards,

Stefan Lötters

Academic Editor

PLOS ONE

Journal Requirements:

3. We note that Figures 1 and 5 in your submission contain map images which may be copyrighted. All PLOS content is published under the Creative Commons Attribution License (CC BY 4.0), which means that the manuscript, images, and Supporting Information files will be freely available online, and any third party is permitted to access, download, copy, distribute, and use these materials in any way, even commercially, with proper attribution. For these reasons, we cannot publish previously copyrighted maps or satellite images created using proprietary data, such as Google software (Google Maps, Street View, and Earth). For more information, see our copyright guidelines: http://journals.plos.org/plosone/s/licenses-and-copyright.

You may seek permission from the original copyright holder of Figures 1 and 5 to publish the content specifically under the CC BY 4.0 license. 

If you are unable to obtain permission from the original copyright holder to publish these figures under the CC BY 4.0 license or if the copyright holder’s requirements are incompatible with the CC BY 4.0 license, please either i) remove the figure or ii) supply a replacement figure that complies with the CC BY 4.0 license. Please check copyright information on all replacement figures and update the figure caption with source information. If applicable, please specify in the figure caption text when a figure is similar but not identical to the original image and is therefore for illustrative purposes only.

Reviewers' comments:

Reviewer's Responses to Questions

**Comments to the Author**

1. Is the manuscript technically sound, and do the data support the conclusions?

Reviewer #1: Yes

2. Has the statistical analysis been performed appropriately and rigorously? 

Reviewer #1: Yes

3. Have the authors made all data underlying the findings in their manuscript fully available?

Reviewer #1: Yes

4. Is the manuscript presented in an intelligible fashion and written in standard English?

Reviewer #1: Yes

5. Review Comments to the Author

Reviewer #1: Impacts of ecotourism and associated human activities on Neotropical wildlife have not been well studies. The current manuscript uses 4 years of camera-trap data to evaluate the occurrence and activity patterns of terrestrial mammals in a nature park in Suriname. Camera traps were located along a gradient of human activity (hikers, vehicles). Results indicated that whereas some species did not appear negatively impacted by tourists, others either tended to avoid areas with more activity or altered their activity patterns to become more nocturnal. The paper is well written, although quite long, methods are clear, and the results well documented. The area is clearly an important area for conservation, with some of the highest capture rates of pumas and jaguars that I have read about. In fact, given that many of the typical prey species do not appear especially abundant (compared to other camera-trap studies), the abundance of the three top felids is remarkable.

Some comments are keyed to line numbers:

Line 2 – patterns

35 – should include scientific names at first mention

108 – are there more recent numbers for tourists?

122-123 – might want to make it clear that you are referring to a diversity index given that species richness was not negatively related to tourist numbers

145-148 – it would useful to know the distribution of rainfall amounts between wet and dry seasons

178 – Figure 1 – the notations on the map are hard to read; also the road to the north is not clear; in contrast, figure 5 clearly shows the roads, the lodging area, and trails; Fig 1 might be clearer if the background was not solid gray and if roads and trails were color coded as in Fig 5

183 – lodging facilities are not apparent to me

201 – what other cameras were used?

280 – remarkably high number of triggers, particularly for the three felids – any explanations for such high numbers? Also, with dual cameras, did you identify individuals for the jaguar and ocelot? Were the triggers mostly from the same individuals walking back and forth along the road?

299 – Figures 2 and 3; if cameras in Figure 1 were numbered then the points in these figures could also be numbered, that would allow readers to see which cameras the dots refer to

305 – make clear that the correlation is between the RAI and number of tourists;

354 – I may have missed it but what are the groups for ANOSIM

358 – Figure 6 – the labels are often overlapping and hard to read

361-366 – seems more like discussion than results

403 -404 – but important to note that there was only one other site in Amazonia (Manaus) and none in the more diverse Neotropical areas closer to the Andes

444 – actually the number of stations along trails (10) was not much less than in this study (16) and at TBS there also were 32 additional stations on two study plots

504 – jaguars are more diurnal, particularly males, in eastern Ecuador

527-529 – but the high rates for large cats suggests lack of an effect

6. PLOS authors have the option to publish the peer review history of their article (what does this mean?). If published, this will include your full peer review and any attached files.

Reviewer #1: No

---

## [Author Response · Author response to Decision Letter 0]

10 May 2021

Response to Reviewers - Impact of ecotourism on abundance, diversity and activity pattern of medium-large terrestrial mammals at Brownsberg Nature Park, Suriname

Comments by academic editor:

Manuscript follows PLOS ONE style.

2. Please review your reference list to ensure that it is complete and correct.

Reference list has been reviewed and the following changes have been made: volume/issue/page additions, journal correction, author corrections, formatting corrections. We added seven references to support our revised manuscript [54-56,76,81-83]. We reviewed all the references at their journal pages and against the retraction watch database (retractiondatabase.org) and found no retracted papers in our list.

3. Regarding the use of copy righted materials in Fig 1 and Fig 5:

We changed the map of South America for one from the CIA The World Factbook 2021. As stated by the CIA "The Factbook is in the public domain. Accordingly, it may be copied freely without permission of the Central Intelligence Agency (CIA)".

The World Factbook 2021. Washington, DC: Central Intelligence Agency, 2021. Political South America.

https://www.cia.gov/the-world-factbook/

The maps of Suriname and the study area were created in Qgis with shapefiles that we own. Roads and trails were mapped by ourselves.

Comments by reviewer #1:

2 – patterns

Added “s”.

35 – should include scientific names at first mention

Included.

108 – are there more recent numbers for tourists?

Unfortunately, STINASU was unable to provide more recent numbers.

122-123 – might want to make it clear that you are referring to a diversity index given that species richness was not negatively related to tourist numbers

Sentence changed.

145-148 – it would useful to know the distribution of rainfall amounts between wet and dry seasons

Data not available for Brownsberg. However, some variation in the dryness of the dry season is reported by De Dijn et al. (2007) and is added to the text.

178 – Figure 1 – the notations on the map are hard to read; also the road to the north is not clear; in contrast, figure 5 clearly shows the roads, the lodging area, and trails; Fig 1 might be clearer if the background was not solid gray and if roads and trails were color coded as in Fig 5

Fig 1 has been fully revised.

183 – lodging facilities are not apparent to me

Tourist facilities indicated in Fig 1 now.

201 – what other cameras were used?

Reconyx PC600 cameras added to the text.

280 – remarkably high number of triggers, particularly for the three felids – any explanations for such high numbers? Also, with dual cameras, did you identify individuals for the jaguar and ocelot? Were the triggers mostly from the same individuals walking back and forth along the road?

Cameras were positioned along roads and trails, consequently producing a high capture rate of the three cats (that prefer using roads to move around). We identified the jaguars to individuals to investigate population parameters. This will be reported in another paper. In the discussion of this manuscript, we mention the density, which is quite high in most periods. Indeed, the resident jaguars triggered the same cameras often. 

299 – Figures 2 and 3; if cameras in Figure 1 were numbered then the points in these figures could also be numbered, that would allow readers to see which cameras the dots refer to

Data points in Fig 2 and 3 labeled with locations.

305 – make clear that the correlation is between the RAI and number of tourists;

It is not the correlation between RAI and number of tourists. It is the correlation between the numbers of a certain mammal species per camera and the number of tourists per camera.

Added a few words in the text.

354 – I may have missed it but what are the groups for ANOSIM

Groups are in the legend at the right of Fig 6.

358 – Figure 6 – the labels are often overlapping and hard to read

Fig 6 has been revised

361-366 – seems more like discussion than results

We think that it is a good introduction/transition to the activity pattern section of the results.

403 -404 – but important to note that there was only one other site in Amazonia (Manaus) and none in the more diverse Neotropical areas closer to the Andes

This is what is published. Are there publications with a higher number of medium-large mammal species from Amazonia or closer to the Andes? If so, we will include these.

444 – actually the number of stations along trails (10) was not much less than in this study (16) and at TBS there also were 32 additional stations on two study plots

We agree that the total effort is probably sufficient to find an effect. We have thus modified our discussion point to the limited timeframe out of a year and added some literature in support of the argument.

504 – jaguars are more diurnal, particularly males, in eastern Ecuador

Text added

527-529 – but the high rates for large cats suggests lack of an effect

We showed an impact on tapir and collared peccaries too. These are important "ecosystem engineers" and their avoidance of areas with many tourists is likely to have an impact on plant composition. The big cats are still present in the area with most disturbance, however have shifted their activity mostly to the night. This is likely to have an effect on both diurnal and nocturnal prey species. We added a part on tapir and peccaries in the discussion.

---

## [Editor Report · Decision Letter 1]

18 May 2021

Impact of ecotourism on abundance, diversity and activity pattern of medium-large terrestrial mammals at Brownsberg Nature Park, Suriname

PONE-D-21-10214R1

Dear Dr. Ouboter,

We’re pleased to inform you that your manuscript has been judged scientifically suitable for publication and will be formally accepted for publication once it meets all outstanding technical requirements.

Kind regards,

Stefan Lötters

Academic Editor

PLOS ONE
---

## [Editor Report · Acceptance letter]

21 May 2021

PONE-D-21-10214R1 

Impact of ecotourism on abundance, diversity and activity patterns of medium-large terrestrial mammals at Brownsberg Nature Park, Suriname 

Dear Dr. Ouboter:

I'm pleased to inform you that your manuscript has been deemed suitable for publication in PLOS ONE. Congratulations! Your manuscript is now with our production department. 

Kind regards, 

on behalf of

Prof. Dr. Stefan Lötters 

Academic Editor

PLOS ONE